# MuLan: Multimodal-LLM Agent for Progressive and Interactive Multi-Object Diffusion

## Abstract

Existing text-to-image models still struggle to generate images of multiple objects, especially in handling their spatial positions, relative sizes, overlapping, and attribute bindings. To efficiently address these challenges, we develop a training-free **Mu**ltimodal-LLM **a**ge**n**t (MuLan), as a human painter, that can progressively generate multi-object with intricate planning and feedback control. MuLan harnesses a large language model (LLM) to decompose a prompt to a sequence of sub-tasks, each generating only one object by stable diffusion, conditioned on previously generated objects. Unlike existing LLM-grounded methods, MuLan only produces a high-level plan at the beginning while the exact size and location of each object are determined upon each sub-task by an LLM and attention guidance. Moreover, MuLan adopts a vision-language model (VLM) to provide feedback to the image generated in each sub-task and control the diffusion model to re-generate the image if it violates the original prompt. Hence, each model in every step of MuLan only needs to address an easy sub-task it is specialized for. The multi-step process also allows human users to monitor the generation process and make preferred changes at any intermediate step via text prompts, thereby improving the human-AI collaboration experience. We collect 200 prompts containing multi-objects with spatial relationships and attribute bindings from different benchmarks to evaluate MuLan. The results demonstrate the superiority of MuLan in generating multiple objects over baselines and its creativity when collaborating with human users.

## 1 Introduction

Diffusion models (Sohl-Dickstein et al., 2015; Ho et al., 2020; Song et al., 2020) have shown growing potential in generative AI tasks, especially in creating diverse and high-quality images with text prompts (Saharia et al., 2022; Rombach et al., 2022). However, current state-of-the-art text-to-image (T2I) models such as Stable Diffusion (Rombach et al., 2022) and DALL-E 3 (Betker et al., 2023) still struggle to deal with complicated prompts involving multiple objects and lack precise control of their spatial relations, potential occlusions, relative sizes, etc. As shown in Figure 2, to generate a sketch of "The orange pumpkin is on the right side of the black door", even the SOTA open-source T2I model, Stable Diffusion XL (Podell et al., 2023), still generates wrong attribute-binding as well as incorrect spatial positions of several objects.

Among works that aim to improve the controllability of T2I models on complicated prompts, a recent promising line of research seeks to utilize large language models (LLMs), e.g., ChatGPT, GPT-4 (Achiam et al., 2023), to guide the generation process (Lian et al., 2023; Feng et al., 2023). Specifically, an LLM is prompted to generate a layout for the given prompt, i.e., a bounding box for each object in the image, given detailed instructions or demonstrations if necessary. However, due to the limited spatial reasoning capability of LLMs as well as their lack of alignment with the diffusion models, it is still challenging for LLMs to directly generate a complete and precise layout for multiple objects. Without a feedback loop interacting with the generative process, the layout's possible mistakes cannot be effectively detected and corrected. Moreover, the layout is often applied as an extra condition in addition to the original prompt (e.g., bounding boxes combined with GLIGEN (Li et al., 2023)), so the diffusion models may still generate an incorrect image due to its misunderstanding of the complicated prompt.

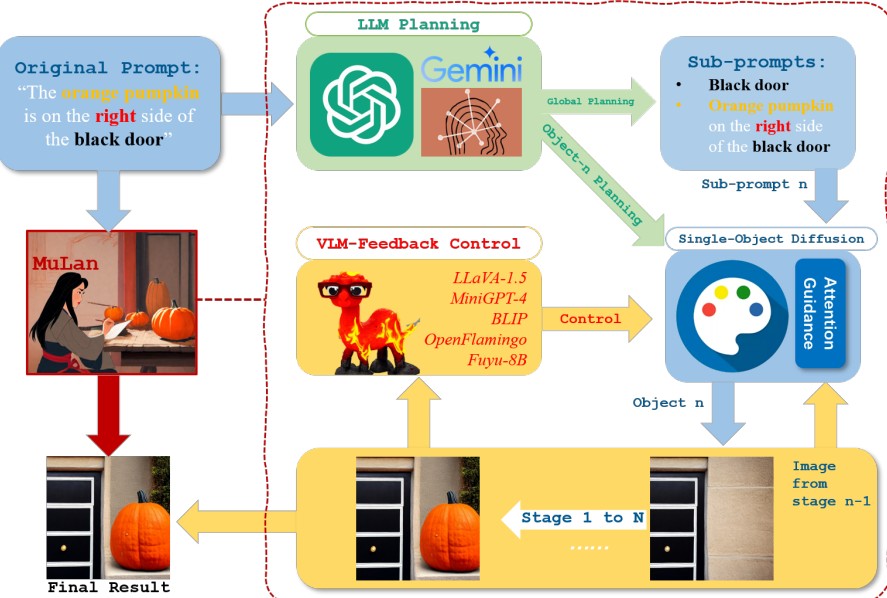

Figure 1: The proposed training-free Multimodal-LLM Agent (MuLan) for Progressive Multi-Object Diffusion. MuLan consists of three main components: (1) LLM planning; (2) Single-object diffusion with attention guidance; and (3) VLM-feedback control. MuLan first decomposes a complicated prompt into a sequence of sub-prompts each for one object, and then generates one object per step conditioned on a sub-prompt and previously generated objects, where LLM plans the rough layout of the object and attention guidance provides an accurate mask for it. The VLM-feedback control allows MuLan to correct mistakes in each step by adjusting hyperparameters in (2).

To address the limitations and challenges of previous methods, we develop a training-free and controllable T2I generation paradigm that does not require demonstrations but mainly focuses on improving the tool usage of existing models. Our paradigm is built upon a progressive multi-object generation by a Multimodal-LLM agent (MuLan), which generates only one object per stage, conditioned on generated objects in the image and attention masks of the most plausible positions to place the new object. Unlike previous methods that add conditions to each model and make the task even more challenging, MuLan uses an LLM as a planner decomposing the original T2I task into a sequence of easier subtasks. Each subtask generates one single object, which can be easily handled by diffusion models. To be noted, the LLM applied at the beginning of MuLan only focuses on high-level planning rather than a precise layout of bounding boxes, while the exact size and position of each object are determined later in each stage by LLM and attention guidance based on the generated objects in the image. Hence, we can avoid mistakes in the planning stage and find a better placement for each object adaptive to the generated content and adhering to the original prompt. In addition, MuLan builds a feedback loop monitoring the generation process, which assesses the generated image per stage using a vision-language model (VLM). When the generated image violates the prompt, the VLM will adjust the diffusion model to re-generate the image so any mistake can be corrected before moving to the next stage. Furthermore, we develop a strategy applied in each stage to handle the overlapping between objects, which is commonly ignored by previous work (Lian et al., 2023).

Therefore, MuLan obtains better controllability of the multi-object composition. An illustration of the progressive generation process is shown in Figure 1. Note that there is a concurrent work called RPG (Yang et al., 2024) sharing a similar high-level idea (i.e., decomposing the prompt into sub-tasks) with MuLan. However, there still exist substantial differences between ours and RPG. MuLan generates each object conditioned on previously generated objects while RPG generates all objects independently. MuLan does not require any manually designed demonstrations for in-context learning. In addition, as shown in Section 4.1, MuLan can be directly applied to human-agent interaction during generation, which greatly boosts the flexibility and effectiveness of the generation. To evaluate MuLan, we curate a dataset of intricate and challenging prompts from

different benchmarks. To compare MuLan with existing approaches, we prompt GPT-4V (OpenAI, 2023) several questions based on the input texts to comprehensively evaluate the alignment of the generated images with the prompts from three aspects. We further conduct human evaluations of the generated images. Extensive experimental results show that MuLan can achieve better controllability over the generation process and generate high-quality images aligning better with the prompts than the baselines. Example images generated by different methods are shown in Figure 2. Our main contributions are summarized as follows:

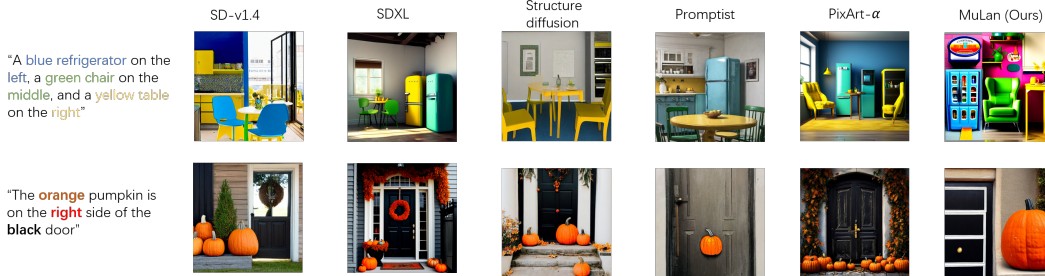

Figure 2: Examples of MuLan-generated images, compared to the original SD-v1.4 (Rombach et al., 2022), the original SDXL (Podell et al., 2023), Structure diffusion (Feng et al., 2022), Promptist (Hao et al., 2022), and PixArt-$\alpha$ (Chen et al., 2023).

- We propose a novel training-free paradigm for text-to-image generation and a Multimodal-LLM agent. It achieves better control in generating images for complicated prompts consisting of multiple objects with specified spatial relationships and attribute bindings.

- We propose an effective strategy to handle multi-object occlusion in T2I generation, which improves the image quality and makes them more realistic.

- We curate a dataset of prompts to evaluate multi-object composition with spatial relationships and attribute bindings in T2I tasks. The quantitative results and human evaluation results show that our method can achieve better results compared to different controllable generation methods and general T2I generation methods.

- We show that the proposed framework can be applied to human-agent interaction during generation. This enables users to effectively monitor and change/adjust the generation process during generation instead of waiting until all the generation is finished.

## 2    RELATED WORK

**Diffusion models**    As a new family of generative models, diffusion models have attracting more and more attention due to its powerful creative capability. Text-to-image generation, which aims to generate the high-quality image aligning with given text prompts, is one of the most popular applications (Nichol et al., 2021; Saharia et al., 2022; Rombach et al., 2022; Betker et al., 2023). Among different powerful diffusion models, the latent diffusion model (Rombach et al., 2022) has shown amazing capability and has been widely used in practice due to the efficiency and superior performance, which is also the backbone of the current SOTA stable diffusion models. Different from the typical diffusion models which directly perform the diffusion and denoising process in the pixel space, the latent diffusion model perform the whole process in the encoded latent space (Rombach et al., 2022), which can greatly reduce the training and inference time. Recently, empowered by a significantly expanded model capacity, Stable Diffusion XL has demonstrated performance levels approaching commercial application standards (Podell et al., 2023). Detailed background on the procedure of diffusion models is provided in Appendix G.

**Composed generation in diffusion models**    Although Stable Diffusion model has shown unprecedented performance on the T2I generation task, it still struggles with text prompts with multi-object, especially when there are several spatial relationships and attribute bindings in the prompts. To achieve more controllable and accurate image compositions, many compositional generation methods have been proposed. StructureDiffusion (Feng et al., 2022) proposed a training-free method to parse the input prompt and combine it with the cross-attention to achieve better control over attribute

bindings and compositional generation. On the other hand, Promptist (Hao et al., 2022) aimed to train a language model with the objective of optimizing input prompts, rendering them more comprehensible and facilitative for diffusion models. Recently, Ranni (Feng et al., 2024) finetunes an LLM to generate bounding boxes and colors. Then they use these as conditions to finetune text-to-image models for image generation. In addition, AnyDoor (Chen et al., 2024b) also requires finetuning of diffusion models for better generation. Several works utilize the large language model to directly generate the whole layout for the input prompt with in-context learning, and then generate the image conditioned on the layout (Lian et al., 2023; Feng et al., 2023; Wu et al., 2024). While all the previous take the whole input prompt, we propose to turn the original complicated task into several easier sub-tasks. A training-free multimodal-LLM agent is utilized to progressively generate objects with feedback control so that the whole generation process would be better controlled. **Very recently, a concurrent work RPG (Yang et al., 2024) also proposed to utilize LLM agent to decompose the prompt into different subtasks. However, MuLan generates each object step by step and correct mistakes after each step rather than treating all subtasks independently and does not need a well-designed in-context learning demonstrations. We defer a more thorough discussion with RPG (Yang et al., 2024) in Appendix B.**

## 3    Multimodal-LLM Agent (MuLan)

Existing diffusion models often struggle with complicated prompts but can handle simpler ones. Recent approaches train a model or apply in-context learning given similar examples to produce a detailed layout for the prompt in advance and the diffusion model can generate each part of the layout with a simpler prompt separately. Rather than generating all objects at once or in parallel, MuLan is inspired by many human painters, who start by making a high-level plan, painting objects one after another as planned, and correcting mistakes after each step if needed. Thereby, the constraints between objects can be naturally taken into account.

### 3.1    Overview

MuLan begins by strategically planning and decomposing an intricate input prompt into a manageable sequence of sub-prompts, each focusing on an easier sub-task generating one single object. MuLan then adopts a progressive strategy that generates one object in each stage conditioned on previously generated objects using a diffusion model. Simultaneously, a VLM offers insightful feedback and adaptively adjusts the generation process to guarantee precision in accomplishing each subtask. Compared to previous methods, MuLan is entirely training-free and does not require any in-context examples. As illustrated in Fig. 1, MuLan is composed of three components:

- **Prompt decomposition by LLM planning**, which produces a sequence of sub-prompts, each focusing on generating one object in the prompt.
- **Conditional single-object diffusion with LLM planning and attention guidance**, which generates a new object conditioned on the previous step's image using a stable diffusion model. While a sub-prompt from LLM planning provides text guidance, the object's size and position are controlled by an attention mask, which guides the object to be correctly positioned and generated.
- **Feedback control by interacting with VLM**, which inspects the image generated per stage and adjusts hyperparameters and attention guidance to re-generate the image if it violates the original prompt.

### 3.2    Prompt Decomposition by LLM Planning

Given a complex prompt $p$, MuLan first uses an LLM to automatically decompose $p$ into $N$ object-wise sub-prompts $p_{1:N}$. During decompostion, MuLan specifically asks the LLM to produce a sequence of objects that will be created in the default order from left to right and bottom to top in the image. The LLM can easily finish this task by leveraging its prior knowledge to fill all objects of $p$ to an empty list of the pre-defined order without in-context learning which requires manually designed examples. Let $\texttt{objs} = \{\texttt{obj}_1, \cdots, \texttt{obj}_n, \cdots, \texttt{obj}_N\}$ be the LLM-planned $N$ objects extracted from $p$. For the first object, the sub-prompt is simply $p_1 =$"$\{\texttt{obj}_1\}$". For object-$n$ with $n > 1$, the subtask is to generate object-$n$ conditioned on previous objects and the textual sub-prompt is defined as $p_n =$"$\{\texttt{obj}_n\}$ and $\{\texttt{obj}_{n-1}\}$". MuLan conducts the above global planning

by an LLM at the very beginning before generating any image. The detailed prompts and template for LLM planning can be found in Appendix I.

When generating each object in Section 3.3, we will use the LLM again as a local planner of the object's position and size, i.e., by generating a mask in the image and coordinating its overlap with previous objects. Then a diffusion model is used to generate the object under the attention guidance of the mask. These will be further elaborated in Section 3.3.

### 3.3 CONDITIONAL SINGLE-OBJECT DIFFUSION WITH LLM PLANNING AND ATTENTION GUIDANCE

At stage-$n$, the diffusion model only focuses on generating $\mathrm{obj}_n$ according to the sub-prompt $\mathrm{p}_n$, ensuring that $\mathrm{obj}_n$ can be correctly positioned and generated. To this end, MuLan utilizes the LLM to plan the relative position and size of $\mathrm{obj}_n$, allocating a rough mask (i.e., a bounding box) $\boldsymbol{M}_n$ for $\mathrm{obj}_n$. Then, cross-attention guidance is applied during the generation of $\mathrm{obj}_n$ to ensure $\mathrm{obj}_n$ is appropriately positioned within $\boldsymbol{M}_n$. The pipeline is given in Figure 3 with the complete procedure listed in Algorithm 1 in Appendix H. We will introduce it step by step in the following.

**LLM Planning of a Rough Mask for $\mathrm{obj}_n$.**
At stage-$n$, MuLan first allocates a rough mask as a bounding box $\boldsymbol{M}_n \triangleq (x_n, y_n, w_n, h_n)$ (x/y coordinates of the top-left corner, width, and height) to guide the generation of $\mathrm{obj}_n$ in the image. As shown in Figure 3, $\boldsymbol{M}_n$ can be derived from $\mathrm{obj}_n$'s relative position $\mathrm{opt}_n \in$ Opts={left,right,top,bottom}, the total number of objects $\mathrm{Num}_n$ in the same position/region as $\mathrm{obj}_n$, and current available space in the image. $\mathrm{Num}_n$ and current available space, combined together, determines the size of $\mathrm{obj}_n$. MuLan utilizes the LLM planner to reason $\mathrm{opt}_n$ and $\mathrm{Num}_n$ given the sub-prompt $\mathrm{p}_n$[1], while the current available space can be determined by the precise mask $\tilde{M}_{n-1}$ which describes the exact position of previously generated $\mathrm{obj}_{n-1}$ and can be easily extracted from the cross-attention maps. It is worth

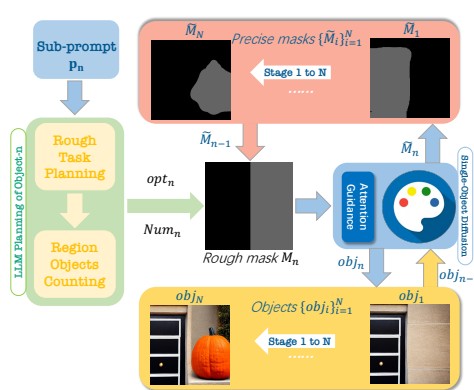

Figure 3: Single object diffusion with LLM planning and attention guidance for $\mathrm{obj}_n$ (detailed procedure in Algorithm 1 in Appendix H).

noting that since there is no previously generated objects for the first object, the available space for $\mathrm{obj}_1$ is the whole image. For detailed computation of $\boldsymbol{M}_n$, please refer to Appendix K.

Once $\boldsymbol{M}_n$ is determined, the cross-attention guidance is utilized during generation of $\mathrm{obj}_n$ to ensure $\mathrm{obj}_n$ is correctly generated within $\boldsymbol{M}_n$, as elaborated in the following.

**Single-Object Generation with Attention Guidance.** Given the rough mask $\boldsymbol{M}_n$ of $\mathrm{obj}_n$, the next is to ensure the generated $\mathrm{obj}_n$ will be correctly located within $\boldsymbol{M}_n$. A natural and intuitive way to achieve this in diffusion models is to guide the generation of the cross-attention map of $\mathrm{obj}_n$, which builds the relevance between the text prompt and the location of generated object.

To this end, MuLan manipulates the cross-attention map of $\mathrm{obj}_n$ under the guidance of $\boldsymbol{M}_n$, using the backward guidance method (Chen et al., 2024a), to maximize the relevance inside $\boldsymbol{M}_n$. Specifically, let $\boldsymbol{A}$ be the cross-attention map, $\boldsymbol{A}_{m,k}$ represents the relevance between the spatial location $m$ and token-$k$ that describes $\mathrm{obj}_n$ in the prompt. Larger value in $\boldsymbol{A}_{m,k}$ indicates that $\mathrm{obj}_n$ is more likely located at the spatial location of $m$. The goal is to maximize the relevance $\boldsymbol{A}_{m,k}$ inside the mask $\boldsymbol{M}_n$ while minimizing the relevance outside the mask $\boldsymbol{M}_n$. Hence the following energy function is utilized:

$$E(\boldsymbol{A},\boldsymbol{M}_n,k)=\left(1-\frac{\sum_{m\in\boldsymbol{M}_n}\boldsymbol{A}_{m,k}}{\sum_m\boldsymbol{A}_{m,k}}\right)^2, \tag{1}$$

---

[1]The detailed prompt template can be found in Appendix J.

where $\sum_{m \in \boldsymbol{M}_n}$ denotes the summation over the spatial locations included in $\boldsymbol{M}_n$, and $\sum_m$ denotes the summation over all the spatial locations in the attention map. In every step-$t$ of the earlier generation process, MuLan applies gradient descent to minimize the energy by updating the input latent $\boldsymbol{z}_{n,t}$ for object $\mathtt{obj}_n$. In this way, the cross-attention map corresponding to $\mathtt{obj}_n$ will achieve the largest relevance inside $\boldsymbol{M}_n$, meaning $\mathtt{obj}_n$ can be correctly positioned inside the rough mask.

On the other hand, to take the previous objects and their constraints into account when generating $\mathtt{obj}_n$, we further combine the latent of $\mathtt{obj}_n$ and $\mathtt{obj}_{n-1}$. Specifically, after step-$t$ of reverse process ($t$ varies from $T$ to 0), we update the latent $\boldsymbol{z}_{n,(t-1)}$ by

$$\boldsymbol{z}_{n,(t-1)} = \boldsymbol{M}'_n \odot \boldsymbol{z}_{n,(t-1)} + (1 - \boldsymbol{M}'_n) \odot \boldsymbol{z}_{(n-1),(t-1)}, \tag{2}$$

where $\odot$ computes element-wise product and $[\boldsymbol{M}'_n]_{uv} = \mathbb{1}_{u \in [x_n, x_n + w_n], v \in [y_n, y_n + h_n]}$ is the 0-1 indicator of whether coordinates $(u, v)$ is included in the bounding box of $\boldsymbol{M}_n$.

MuLan applies the above single-object diffusion to each object one after another from $\mathtt{obj}_1$ to $\mathtt{obj}_N$, as planned by the LLM at the very beginning. The procedure of generating $\mathtt{obj}_n$ is detailed in Algorithm 1.

**Objects Overlapping.** Overlapping between objects is a key challenge in text-to-image diffusion models. However, it lacks attention in previous methods (Lian et al., 2023; Feng et al., 2023). Instead, we propose an effective strategy that can be merged into the procedure above. Specifically, at the generation of object $\mathtt{obj}_n$, we prompt the LLM to judge if there is overlapping between $\mathtt{obj}_n$ and $\mathtt{obj}_{n-1}$. If there is overlapping, we first compute three candidates for the rough mask $\{\boldsymbol{M}_{n,i}\}_{i=1}^3$, associated with three overlapping ratios $\{r_i\}_{i=1}^3 = \{10\%, 30\%, 50\%\}$ between $\mathtt{obj}_{n-1}$ and $\mathtt{obj}_n$.

Given the three masks $\boldsymbol{M}_{n,i}$, MuLan generates three candidate images using Algorithm 1. Then the CLIP scores (Hessel et al., 2021) between the generated images and the input prompt $\mathtt{p}_n$ are computed and the image with the maximal CLIP score is selected as the generated image for $\mathtt{obj}_n$. An illustration is given in Figure 11 with more details of candidate masks in Appendix L.

## 3.4 Interaction with VLM and Human Users during Generation

To correct the possible mistakes made in the sequential generation process, MuLan builds an adaptive feedback-loop control by interacting with a vision-language model (VLM). After each generation stage, MuLan queries the VLM to inspect the generated object(s) and its consistency with the input prompt. If they do not align well, MuLan will adjust the backward guidance of the current stage to re-generate the object. More specifically, MuLan will modify the hyperparameters of backward guidance to control the strength of the guidance. We empirically found that the errors are typically the size or the position of the generated object. For example, the object may be too large and outside the rough mask. Hence the guidance strength needs to be larger to make the object smaller. In the whole generation process, if MuLan needs to regenerate an object, it will try different guidance strength, i.e., the weight of the gradient of the energy function (Eq. 1), and the loss threshold that is used for stopping criteria of guidance. In cases with incorrect positions, it will also re-plan the spatial location and regenerate the object. Such a close-loop control involves LLM, diffusion, and VLM and significantly automates the T2I generation for complicated prompts, leading to a more accurate generation in practice.

In addition, the multi-step process naturally allows human-agent interaction/collaboration during generation in practice. Users can timely monitor the generation process. In this way, the interaction enables users to make preferred changes and adjustments to the generated images easily and effectively by providing adjusting prompts to MuLan at any intermediate step, such as attribute adjustment, object adjustment, and spatial relationship adjustment. With the adjusting prompts, MuLan will utilize the LLM to modify the original prompt accordingly and change the generation process to the preferred one. An illustration for different changes or adjustments during generation is shown in Figure 4, which indicates MuLan can achieve both simple and composed complex adjustments with interaction. In contrast, for other existing generation and editing methods, users have to wait until the whole generation process is finished. Therefore, the proposed framework is more user-friendly and flexible in terms of human-agent interaction and collaboration.

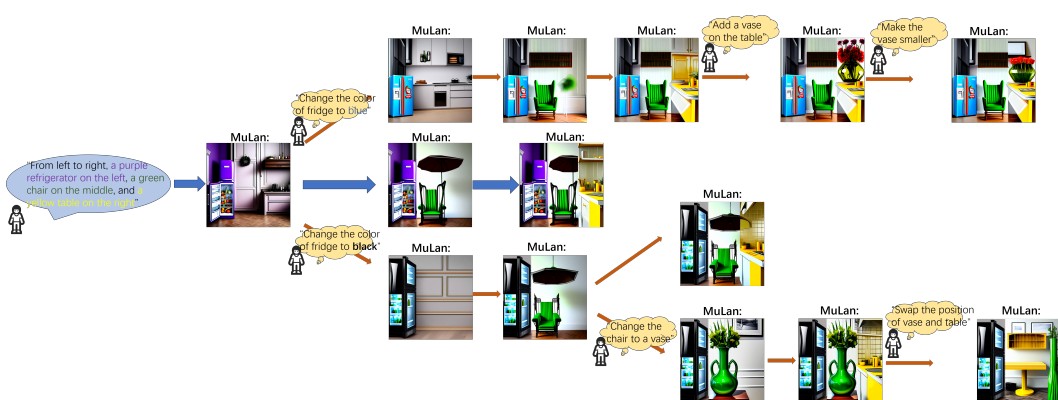

Figure 4: An illustration tree for difference cases of **human-agent interaction during generation**. The middle branch (connected by blue arrows) shows the original generation process without human-agent interaction. The top and bottom branches show different complex composed human-agent interaction during generation for various adjustments, involving object adjustments, attribute adjustments, and spatial relationship adjustments, which demonstrate the flexibility and effectiveness of MuLan for human-agent interaction during generation.

## 4 EXPERIMENTS

**Dataset**   To evaluate our framework, we construct a prompt dataset from different benchmarks. Specifically, since our focus is to achieve better generation for complex prompts containing multi-objects with both spatial relationships and attribute bindings, we first collect all complex spatial prompts from T2I-CompBench (Huang et al., 2023). To make the experiments more comprehensive, we let ChatGPT generate about 400 prompts with different objects, spatial relationships, and attribute bindings so that the prompt sets consists of about 600 prompts. To further evaluate the capability of our framework on extremely complex and hard prompts, we manually add prompts that SDXL fails to generate, leading to a hard prompt dataset containing 200 prompts. Similar to the complex spatial prompts in T2I-CompBench (Huang et al., 2023), each prompt in our curated dataset typically contains two objects with various spatial relationships, with each object containing attribute bindings randomly selected from {color, shape, texture}.

**Models & Baseline**   As a training-free framework, MuLan can be incorporated into any existing diffusion models. We evaluate two stable diffusion models with our framework, Stable Diffusion v1.4 (Rombach et al., 2022) and the SOTA Stable Diffusion XL (Podell et al., 2023). To verify the superiority of MuLan, we compare it with previous controllable generation methods and general T2I generation methods. Specifically, we evaluate Structure Diffusion (Feng et al., 2022), Prompt-tist (Hao et al., 2022), the original Stable Diffusion v1.4, the original SDXL, and the recent SOTA diffusion model PixArt-$\alpha$ (Chen et al., 2023).

**Implementation Details**   MuLan use GPT-4 (Achiam et al., 2023) as the LLM planner, and LLaVA-1.5 (Liu et al., 2023) as the VLM checker to provide the feedback. We also conducted an ablation study to show the importance of the feedback control provided by the VLM and the effect of different VLMs. Moreover, we found the attention blocks utilized during the attention guidance are vital, which can be classified as near-input blocks, near-middle blocks, and near-output blocks. We utilize the near-middle blocks in our main experiments and also show the ablation results of different block. Our codes (including the prompt dataset) are available in the supplementary material. All the experiments are conducted on a single NVIDIA RTX A6000 GPU.

**Evaluation**   Since the prompt dataset contains texts with complex compositions, we design a questionnaire to comprehensively investigate the alignment between the generated image and the corresponding input text. The questionnaire is composed of three aspects - object completeness, correctness of attribute bindings, and correctness of spatial relationships. We only set two options for each

question (Yes or No), without any ambiguity. For detailed questions and examples of the evaluation, please refer to Appendix M. For each aspect of the evaluation, we compute the percentage of answers with "Yes". Given the generated image, we assess the image's quality using a questionnaire asking both the state-of-the-art multi-modal large language model (GPT-4V (OpenAI, 2023)) and the human evaluator.

## 4.1 MAIN RESULTS AND ANALYSIS

**Results on GPT Evaluation**   Given the generated image, we prompt GPT-4V to answer the questions about the image in the questionnaire, where each only focuses on one of the three aspects. The results for different methods and different base models are shown in Table 1. The results show that our framework can achieve the best performance compared to different controllable generation methods and T2I generation methods. In particular, in the two 'harder' aspects - attribute bindings and spatial relationships, MuLan can surpass other methods by a large margin. More results can be found in Figure 5 and Appendix O.

Table 1: **GPT-4V evaluation**/**human evaluation** of images generated by different methods for complicated prompts.

| Method | Object completeness | Attribute bindings | Spatial relationships | Overall |
|---|---|---|---|---|
| Structure Diffusion (Feng et al., 2022) | 88.97%/87.37% | 54.62%/62.63% | 34.36%/24.24% | 64.31%/64.85% |
| Promptist-SD v1.4 (Hao et al., 2022) | 80.36%/70.71% | 49.23%/52.02% | 24.49%/13.13% | 56.73%/51.72% |
| Promptist-SDXL (Hao et al., 2022) | 94.36%/**93.94%** | 70.00%/78.28% | 35.89%/33.33% | 72.92%/75.56% |
| SD v1.4 (Rombach et al., 2022) | 90.31%/74.49% | 57.14%/51.02% | 37.24%/32.65% | 66.43%/56.73% |
| SDXL (Podell et al., 2023) | 94.64%/78.57% | 66.07%/53.06% | 41.14%/24.49% | 72.34%/57.55% |
| PixArt-$\alpha$ (Chen et al., 2023) | 92.09%/76.53% | 66.58%/61.22% | 34.69%/32.65% | 70.41%/61.63% |
| **MuLan-SD v1.4 (Ours)** | 93.11%/86.36% | 74.23%/74.24% | **51.53%**/**54.54%** | **77.24%**/75.15% |
| **MuLan-SDXL (Ours)** | **96.17%**/90.40% | **75.00%**/**79.29%** | 39.29%/49.49% | 76.33%/**77.78%** |

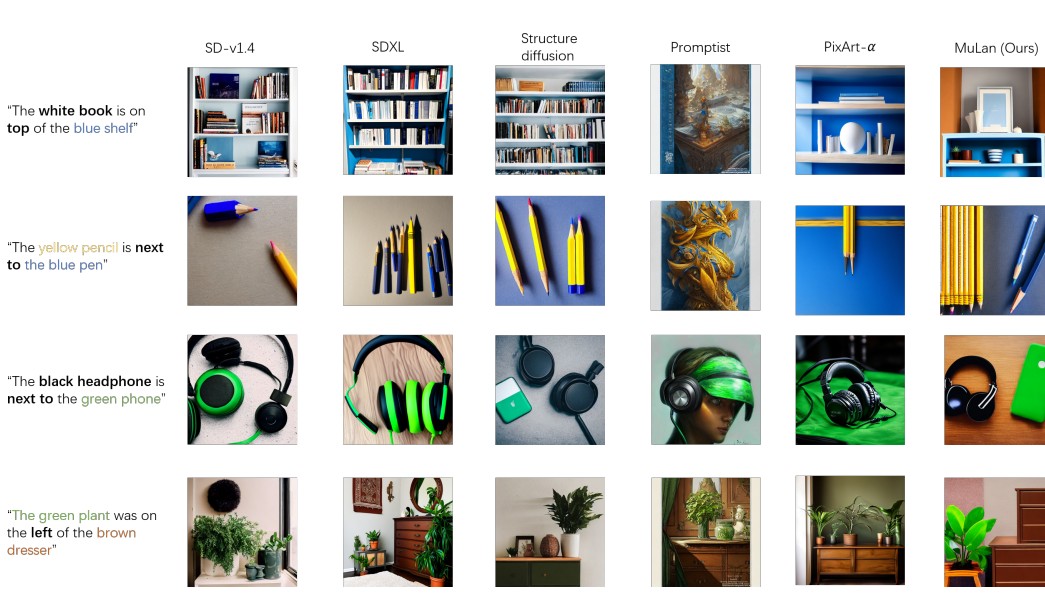

Figure 5: More qualitative examples of images generated by different methods on intricate prompts.

**Results on Human Evaluation**   To further accurately evaluate the generated images about the alignments with human preferences, we further conduct a human evaluation by randomly sampling 100 prompts from the prompt dataset. Similarly, we ask human evaluators to finish the questionnaire used in GPT evaluation. The results are shown in Table 1, which indicates that our method can still achieve the best performance and is consistent with the GPT-4V evaluation results.

**Results on Human-Agent Interaction**   To show MuLan is still very effective if users want to modify the input prompt or edit the generated images during the generation, i.e., the human-agent

interaction, we use ChatGPT to mimic the user to generate various adjusting prompts for the interaction with MuLan on randomly sampled 50 prompts. SD v1.4 (Rombach et al., 2022) is utilized as the base model. The generated adjusting prompts focus on several aspect, i.e., attribute adjustment, object adjustment, and spatial relationship adjustment. We use GPT-4V (OpenAI, 2023) to quantitatively evaluate the performance of MuLan given the final generated images and final text prompts, as shown in Table 2. The results indicate that MuLan can still achieve high accuracy even with various adjustments/changes during generation.

Table 2: GPT-4V evaluation of final generated images and final prompts after adjustments/changes. The results show that MuLan is still very effective with various adjustment of prompts during generation.

|  | Objects | Attributes | Spatial | Overall |
|---|---|---|---|---|
| MuLan-SD v1.4 | 95.92% | 72.45% | 28.57% | 73.06% |

## 4.2 ABLATION STUDY

In this section, we show ablation results on the effect of the attention blocks during diffusion generation and the importance of the VLM feedback control in the proposed framework. 50 prompts are randomly sampled from the prompt dataset for all experiments in the ablation study.

**Ablation on the attention blocks**   As we mentioned at the beginning of Section 4, there are three options for the attention blocks used for backward guidance, i.e., near-input blocks, near-middle blocks, and near-output blocks. We empirically found the near-middle blocks can achieve the best control and performance for the generation, which generally contains the richest semantics. Hence here we show the ablation results on different choices of the attention blocks. We utilize SD-v1.4 as the base model, and evaluate the performance of different attention blocks under our framework by GPT-4V. The results are shown in Table 3, which indicates the diffusion generation with near-middle blocks can achieve much better results compared to the other two options.

Table 3: **Ablation study on attention blocks** with SD-v1.4 as the base model. "Objects", "Attributes", and "Spatial" denote Object completeness, Attribute bindings, and Spatial relationships. The results (evaluated by GPT-4V (OpenAI, 2023)) show that near-middle attention blocks perform the best for attention guidance.

| Guidance | Objects | Attributes | Spatial | Overall |
|---|---|---|---|---|
| near-input | 83.67% | 55.10% | 14.29% | 58.37% |
| near-middle | **97.96%** | **80.61%** | **30.61%** | **77.55%** |
| near-output | 72.45% | 45.92% | 22.45% | 51.84% |

**Ablation on the VLM feedback control**   The VLM feedback control is a key componenet in MuLan to provide feedback and adjust the generation process to ensure the every stage's correct generation. Here, we show the importance of the feedback by removing feedback control from the whole framework. As shown in Table 4, after removing the VLM, the results would be much worse. It is because there is no guarantee or adaptive adjustment for each generation stage, which verifies that the feedback control provided by the VLM is essential to handle complex prompts. Moreover, we also test MuLan's compatibility with different VLMs. As shown in Table 5, we compare the Mulan's performance using different VLMs including LLaVA-1.5 (Liu et al., 2023), GPT-4V (OpenAI, 2023), and Gemini-Pro (Team et al., 2023). The results show that MuLan could still maintain a good performance with different choices of the VLM and achieve good compatibility.

Table 4: **Ablation study** comparing **MuLan with vs. without VLM feedback** control, using SD-v1.4 as the diffusion model and GPT-4 as the judge in evaluations. It indicates that feedback control can significantly improve the performance.

| MuLan | Objects | Attributes | Spatial | Overall |
|---|---|---|---|---|
| w/ Feedback | **97.96%** | **80.61%** | **30.61%** | **77.55%** |
| w/o Feedback | 81.63% | 59.18% | 18.37% | 60.00% |

Table 5: **Ablation study of the VLM** used in MuLan, using SD-v1.4 as the diffusion model and GPT-4 as the judge in evaluations. The results show that the choice of the VLM would not affect the overall performance too much.

| VLM in MuLan | Objects | Attributes | Spatial | Overall |
|---|---|---|---|---|
| LLaVA-1.5 (Liu et al., 2023) | 97.96% | 80.61% | 30.61% | 77.55% |
| GPT-4V (OpenAI, 2023) | 95.92% | 80.61% | 28.57% | 76.33% |
| Gemini-Pro (Team et al., 2023) | 95.92% | 83.67% | 38.78% | 79.59% |

## 5 CONCLUSIONS AND LIMITATIONS

In this paper, we propose a training-free multimodal-LLM agent (MuLan) to progressively generate objects contained in the complicated input prompt with closed-loop feedback control, achieving better and more precise control on the whole generation process. By first decomposing the complicated prompt into easier sub-tasks, our method takes turns to deal with each object, conditioned on the previous one. The VLM checker further provides a guarantee with feedback control and adaptive adjustment for correct generation at each stage. Moreover, the application to the human-agent interaction further enhances the significance of MuLan, making the generation more flexible and effective to align with the preferences of users. Extensive experiments demonstrate the superiority of MuLan over previous methods, showing the potential of MuLan as a new paradigm of controllable diffusion generation. However, there are still limitations to be further addressed in the future work. Since the whole generation contains multiple stages, depending on the number of objects, it will take a longer time than a one-stage generation approach. On the other hand, MuLan may also fail to generate correct objects in some non-common corner cases of image composition. We defer more detailed discussion and illustrations of the limitations to Appendix N.

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

## A BROADER IMPACT

Our work will bring significant advantages to both the research community focused on diffusion models and the practical application of T2I generation.

In terms of the research community, we present a new and novel controllable image generation paradigm that demonstrates exceptional controllability and produces remarkable results even when tackling challenging tasks. This pioneering approach can offer valuable insights for future investigations into diffusion models.

Regarding industrial applications, our method can be readily employed by T2I generation service providers to enhance the performance of their models. Moreover, the diffusion models operating within our framework are less likely to generate harmful content due to the meticulous control exerted at each generation stage.

## B DIFFERENCES BETWEEN MULAN AND THE CONCURRENT WORK RPG

As stated in Introduction and Related work, although we acknowledge that our proposed framework shares a similar high-level idea with RPG, we would like to emphasize that there are still substantial differences between ours and RPG.

Firstly, our proposed MuLan aims to progressively generate each object given each subprompt. At the same time, the objects are generated conditioned on previously generated objects. In RPG, on the other hand, all objects are generated independently. In addition, different from RPG which requires manually designed in-context examples for the CoT reasoning, ours does not have such requirement. We directly utilize LLMs for the planning during generation, which is an easier task and can be done by LLMs without in-context learning. What's more, MuLan can adaptively control and correct the generation results using feedback by the VLMs while RPG does not have the feedback for the generation. Also, for the common overlapping problem between objects, we propose a strategy to generate several candidates to deal with it. In contrast, in RPG, the overlapping parts are treated as a whole for generation.

More importantly, as we show in Section 4.1, our proposed framework can be directly applied to human-agent interaction during generation to facilitate flexible and effective changes/adjustments of the process while RPG cannot achieve the interaction. To summarize, the main differences between MuLan and RPG are as follows:

- Our proposed MuLan generates each object conditioned on previously generated objects while RPG generates all objects in parallel independently.
- MuLan does not require any in-context learning during the whole generation; in RPG, specifically designed in-context examples are needed for Chain-of-Thought reasoning.
- MuLan utilizes the VLM-based feedback control to ensure each object can be generated correctly while RPG does NOT have such a feedback mechanism.
- We propose a strategy to deal with overlapping/interaction between objects whereas RPG directly treats overlapping objects as a whole part to generate.
- MuLan can be directly applied to human-agent interaction during generation for flexible and various adjustments of the generation process while RPG cannot achieve it.

## C MORE COMPARISON RESULTS WITH CONTROLLABLE IMAGE GENERATION METHODS

Here we present more quantitative results between MuLan and other state-of-the-art controllable image generation methods, Ranni (Feng et al., 2024) and Composable Diffusion (Liu et al., 2022). We randomly sample 50 prompts from the prompt dataset and use GPT-4V to evaluate the alignment between generated images and prompts.

The results are shown in Table 6, indicating that MuLan is much better and even outperforms training-based controllable generation mthods.

Table 6: GPT-4V evaluation of MuLan and more controllable generation methods. The results show that MuLan with SD-v1.4 performs better, even surpassing training-based methods.

|  | Objects | Attributes | Spatial | Overall |
|---|---|---|---|---|
| MuLan-SD v1.4 | 97.96% | 80.61% | 30.61% | 77.55% |
| Ranni (Feng et al., 2024) | 70.41% | 38.78% | 20.41% | 47.76% |
| Composable Diffusion (Liu et al., 2022) | 90.82% | 63.27% | 22.45% | 66.12% |

## D    COMPARISON WITH STABLE DIFFUSION 3

To further evaluate the effectiveness of the proposed training-free framework MuLan, we also qualitatively compare MuLan with the latest state-of-the-art text-to-image generation model, Stable Diffusion 3 (Esser et al., 2024). As shown in Figure 6, even Stable Diffusion 3 cannot deal with prompts with simple spatial relationships steadily, while MuLan with SD-v1.4 can achieve controllable generation and generate correct images that align with prompts, indicating the effectiveness of the proposed framework.

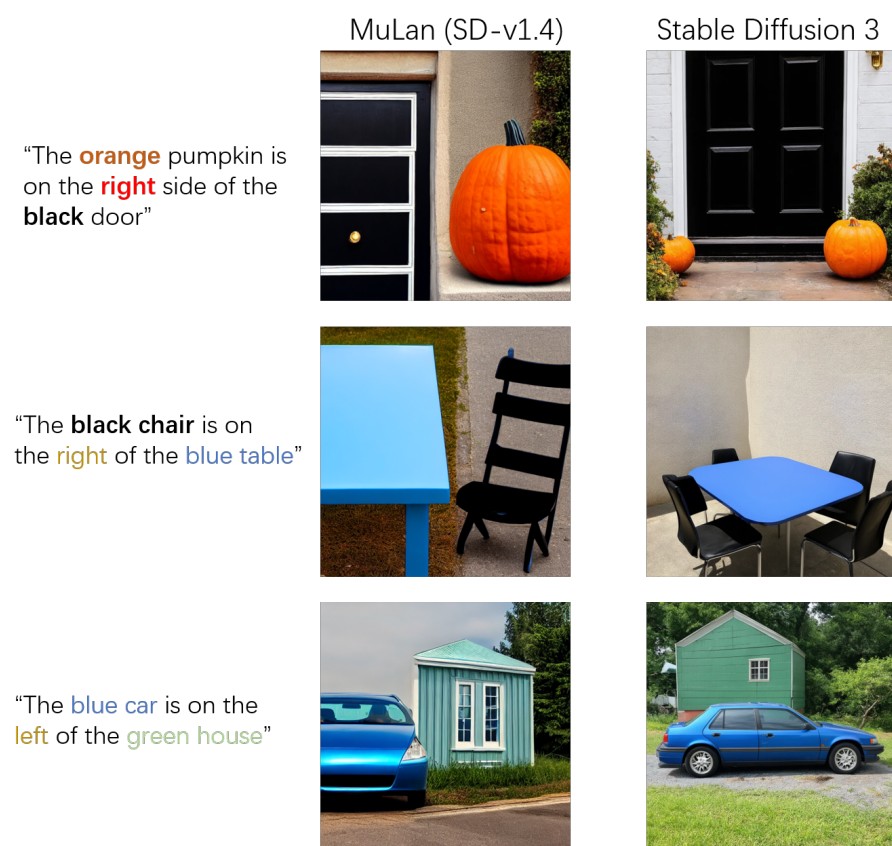

Figure 6: Qualitative comparison between MuLan and Stable Diffusion 3.

## E    VISUAL QUALITY AND REALISM OF MULAN-GENERATED IMAGES

Please note that since MuLan is training-free, the visual quality and realism of generated images highly depend on the utilized base models, e.g., SD v-1.4, SDXL, etc. MuLan does not degrade the visual quality of generated images. To further show this, we present more visualization results of MuLan and the base models. As shown in Figure 7, MuLan with SDXL and the original SDXL have very similar performance in terms of visual quality and realism.

MuLan with SDXL          Original SDXL

"The red umbrella is on **top** of the white coat rack"

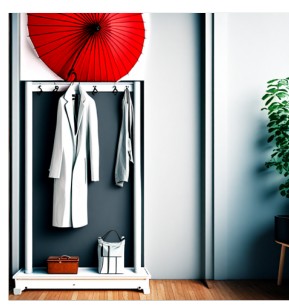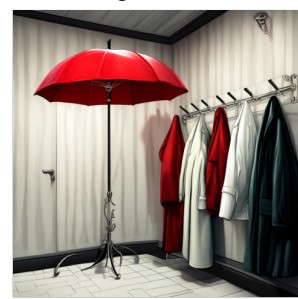

"The green plant is next to the blue vase"

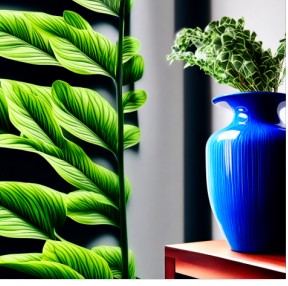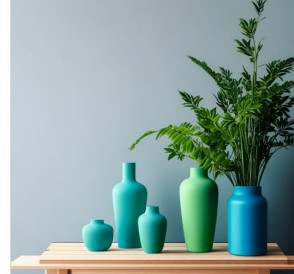

"The orange balloon is on **top** of the yellow box"

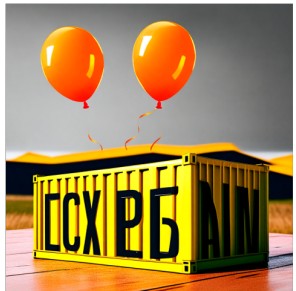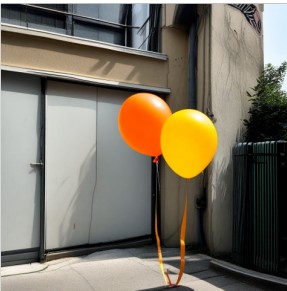

Figure 7: Visual quality and realism comparison between MuLan and the original base model.

## F   MORE RESULTS ON COMPLEX OVERLAPPING PROMPTS

To further verify the effectiveness of the proposed overlapping processing module, we show more visualization results on complex overlapping prompts, including interaction between animals and humans. As shown in Figure 11, MuLan can deal with complex overlapping prompts better and show effectiveness for different overlapping cases.

Figure 8: Visualization results on complex overlapping prompts.

## G   BACKGROUND ON (LATENT) DIFFUSION MODELS

Consisting of the diffusion process and the reverse process, diffusion models have shown impressive capability for high-quality image generation by iteratively adding noise and denoising (Ho et al., 2020). Let $x_0 \sim q(x_0)$ be the true data distribution. Starting from $x_0$, the diffusion process adds different levels of noise pre-defined by the schedule $\{\beta_t\}_1^T$, producing $x_1, \cdots, x_T$. As $T \to \infty$, $x_T$ will become the standard Gaussian distribution $\mathcal{N}(\mathbf{0}, \mathbf{I})$. Accordingly, the reverse process aims to reverse the above process and reconstruct the true data distribution from $p(x_T) = \mathcal{N}(\mathbf{0}, \mathbf{I})$ by a parameterized noise model $\epsilon_\theta(\cdot)$. With $\boldsymbol{\epsilon} \sim \mathcal{N}(\mathbf{0}, \mathbf{I})$, the training loss of the model can be simplified as

$$L(\theta) = \mathbb{E}_{t, x_0, \boldsymbol{\epsilon}} \| \boldsymbol{\epsilon} - \boldsymbol{\epsilon}_\theta(\sqrt{\bar{\alpha}_t} x_0 + \sqrt{1 - \bar{\alpha}_t} \boldsymbol{\epsilon}, t) \|^2. \tag{3}$$

Latent diffusion models (Rombach et al., 2022) have recently attracted growing attention due to their efficiency and superior performance. Instead of performing diffusion and its reverse process in the pixel space, they add noise and denoise in a latent space of $z$ encoded by a pre-trained encoder $\mathcal{E}$. Thereby, the diffusion process starts from $z_0 = \mathcal{E}(x_0)$ and subsequently produces latent states $z_1, \cdots, z_t, \cdots, z_T$. Accordingly, the training loss becomes

$$L_{LDM} = \mathbb{E}_{z_0, \boldsymbol{\epsilon}, t} \| \boldsymbol{\epsilon} - \boldsymbol{\epsilon}_\theta(z_t, t) \|^2. \tag{4}$$

## H   ALGORITHM PROCEDURE OF SINGLE-OBJECT DIFFUSION IN MULAN

The complete and detailed procedure of single object diffusion described in Section 3.3 is shown in Algorithm 1.

---

**Algorithm 1** Single Object Diffusion in MuLan

---

1: **Input:** Object number $n$, sub-prompt $\mathtt{p}_n$, LLM planner $\mathtt{Planner}$, precise mask $\tilde{\boldsymbol{M}}_{n-1}$ (only for $n > 1$), latents $\{\boldsymbol{z}_{(n-1),(t-1)}\}_{t=1}^{T}$ (only for $n > 1$), attention guidance timestep threshold $T'$, combination timestep threshold $T^*$ (only for $n > 1$), learning rate $\eta$, diffusion model $\mathcal{D}$.
2: **Output:** Image with $\mathtt{obj}_n$ and its precise mask $\tilde{\boldsymbol{M}}_n$.
3: **if** $n = 1$ **then**
4:     $\mathtt{opt}_1, \mathtt{Num}_1 = \mathtt{Planner}(\mathtt{p}_1)$
5:     Apply Eq. equation 5 to compute $\boldsymbol{M}_1$
6:     **for** $t = T, \cdots, 1$ **do**
7:         **if** $t > T'$ **then**
8:             $\boldsymbol{z}_{1,t} = \boldsymbol{z}_{1,t} - \eta \cdot \nabla_{\boldsymbol{z}_{1,t}} E(\boldsymbol{A}, \boldsymbol{M}_1, k)$
9:         **end if**
10:        $\boldsymbol{z}_{1,(t-1)} = \mathcal{D}(\boldsymbol{z}_{1,t}, t, \mathtt{p}_1)$ {Single denoising step}
11:    **end for**
12: **else**
13:    $\mathtt{opt}_n, \mathtt{Num}_n = \mathtt{Planner}(\mathtt{p}_n, \{\mathtt{obj}_i\}_{i=1}^{n-1})$
14:    Apply Eq. equation 6 to compute $\boldsymbol{M}_n$
15:    **for** $t = T, \cdots, 1$ **do**
16:        **if** $t > T'$ **then**
17:            $\boldsymbol{z}_{n,t} = \boldsymbol{z}_{n,t} - \eta \cdot \nabla_{\boldsymbol{z}_{n,t}} E(\boldsymbol{A}, \boldsymbol{M}_n, k)$
18:        **end if**
19:        $\boldsymbol{z}_{n,(t-1)} = \mathcal{D}(\boldsymbol{z}_{n,t}, t, \mathtt{p}_n)$
20:        **if** $t > T^*$ **then**
21:            Apply Eq. equation 2 to combine latent of $\mathtt{obj}_n$ and $\mathtt{obj}_{n-1}$
22:        **end if**
23:    **end for**
24: **end if**
25: $\mathtt{obj}_n = \boldsymbol{z}_{n,0}$
26: $\tilde{\boldsymbol{M}}_n = (\tilde{x}_n, \tilde{y}_n, \tilde{w}_n, \tilde{h}_n)$, a bounding box based on thresholding of $\frac{1}{|B|} \sum_{j \in B} \boldsymbol{A}_{(:,k)}^{(j)}$ {Token-$k$ corresponds to $\mathtt{obj}_n$}

---

# I    DETAILED PROMPT TEMPLATE OF THE GLOBAL PLANNING BY THE LLM

As stated in Section 3.2, MuLan first conduct the global planning to decompose the input prompts into $N$ objects before the whole generation process. To this end, given the input prompt $\mathtt{p}$, we prompt the LLM using the following template:

> You are an excellent painter. I will give you some descriptions. Your task is to turn the description into a painting. You only need to list the objects in the description by painting order, from left to right, from down to top. Do not list additional information other than the objects mentioned in the description. Description: $\{\mathtt{p}\}$.

In this way, the LLM will decompose the input prompt $\mathtt{p}$ following the pre-defined order.

# J    DETAILED PROMPT TEMPLATE OF THE LOCAL PLANNING BY THE LLM

As stated in Section 3.3, the LLM is also utilized during the generation stage for local planning of the object's rough position and the object counting.

For the rough position $\mathtt{opt}_1$ planning of the first object, we utilize the following template:

> You are an excellent painter. I will give you some descriptions. Your task is to turn the description into a painting. Now given the description: $\{\mathtt{p}\}$. If I want to paint the $\{\mathtt{obj}_1\}$ in the painting firstly, where to put the $\{\mathtt{obj}_1\}$? Choose from left, right, top, and bottom. You can make reasonable guesses. Give one answer.

Then the LLM is prompted to figure out the object number based on $\mathtt{opt}_1$.

If $\mathtt{opt}_1 = \mathtt{left}$, the prompt template for $\mathtt{obj}_1$ is:

> You are an excellent painter. I will give you some descriptions. Your task is to turn the description into a painting. Now given the description: {p}. How many non-overlapping objects are there in the horizontal direction? ONLY give the final number.

If $\text{opt}_1 = \text{bottom}$, the prompt template would be:

> You are an excellent painter. I will give you some descriptions. Your task is to turn the description into a painting. Now given the description: {p}. How many non-overlapping objects are there in the vertical direction? ONLY give the final number.

For the rough position $\text{opt}_n (n \geq 2)$, we utilize the following template:

> You are an excellent painter. I will give you some descriptions. Your task is to turn the description into a painting. Now given the description: {p}. If I already have a painting that contains $\{\{\text{obj}_i\}_{i=1}^{n-1}\}$, what is the position of the $\{\text{obj}_n\}$ relative to the $\{\text{obj}_{n-1}\}$? Choose from left, right, above, bottom, and none of above. You can make reasonable guesses. Give one answer.

Then we prompt the LLM to figure out the object number by:

> You are an excellent painter. I will give you some descriptions. Your task is to turn the description into a painting. Now given the description: {p}. If I already have a painting that contains $\{\{\text{obj}_i\}_{i=1}^{n-1}\}$, how many objects are there in/on the $\{\text{opt}_n\}$ of $\{\text{obj}_{n-1}\}$? Only give the final number.

## K    DETAILS FOR THE COMPUTATION OF ROUGH MASKS

**When** $n = 1$, since there is no object generated yet, both the position $\text{opt}_1$ and $\text{Num}_1$ are unrestricted and the LLM can be prompted to determine $\text{opt}_1$ and $\text{Num}_1$ given sub-prompt $p_1$. Since the object order starts from left to right and bottom to top, there will be only two position options $\text{opt}_1 \in \{\text{left}, \text{bottom}\}$ for $\text{obj}_1$. Once $\text{opt}_1$ determined, MuLan evenly splits the whole image's width/height ($W/H$) to $\text{Num}_1$ parts and assigns the very left (bottom) part to $\text{obj}_1$, which leads to the following bounding box (an illustration for the computation is shown in Figure 9):

$$M_1 = \begin{cases} (0, 0, \frac{W}{\text{Num}_1}, H), & \text{if } \text{opt}_1 = \text{left}, \\ (\frac{(\text{Num}_1 - 1) \cdot H}{\text{Num}_1}, 0, W, \frac{H}{\text{Num}_1}), & \text{if } \text{opt}_1 = \text{bottom}. \end{cases} \tag{5}$$

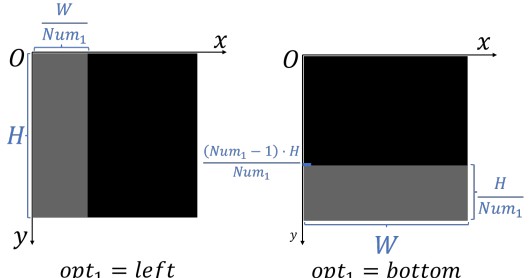

Figure 9: Illustration of the rough mask $M_1$ of $\text{obj}_1$. There are only two options left, bottom for the mask since the LLM is prompted to plan the object order from left to right, bottom to top.

**When** $n > 1$, the position $\text{opt}_n$ denotes $\{\text{obj}\}_n$'s relational position to the previous object $\{\text{obj}\}_{n-1}$. Since MuLan generates objects from left to right and from bottom to top, $\text{opt}_n \in \{\text{right}, \text{top}\}$. Given sub-prompt $p_n$, an LLM is prompted to select $\text{opt}_n$ and determine $\text{Num}_n$. Meanwhile, the precise mask $\tilde{M}_{n-1} = (\tilde{x}_{n-1}, \tilde{y}_{n-1}, \tilde{w}_{n-1}, \tilde{h}_{n-1})$ of $\text{opt}_{n-1}$ can be extracted from the image with $\{\text{obj}\}_{n-1}$ generated (e.g., by text-image cross-attention maps in the diffusion model), which is utilized as the condition for the computation of bounding box boundary of the rough mask $M_n$. Hence, the rough mask $M_n$ for $\text{obj}_n$ can be derived from $\text{opt}_n$, $\text{Num}_n$, and

$\tilde{M}_{n-1}$ as followings.

$$M_n = \begin{cases} \left(\tilde{x}_{n-1} + \tilde{w}_{n-1}, 0, \frac{W - \tilde{x}_{n-1} + \tilde{w}_{n-1}}{\text{Num}_n}, H\right), & \text{if } \text{opt}_n = \text{right}, \\ \left(0, \frac{\tilde{y}_{n-1} \cdot (\text{Num}_n - 1)}{\text{Num}_n}, W, \frac{\tilde{y}_{n-1}}{\text{Num}_n}\right), & \text{if } \text{opt}_n = \text{top}. \end{cases} \quad (6)$$

Figure 10 illustrates how the rough mask can be computed based on the precise mask of previous objects.

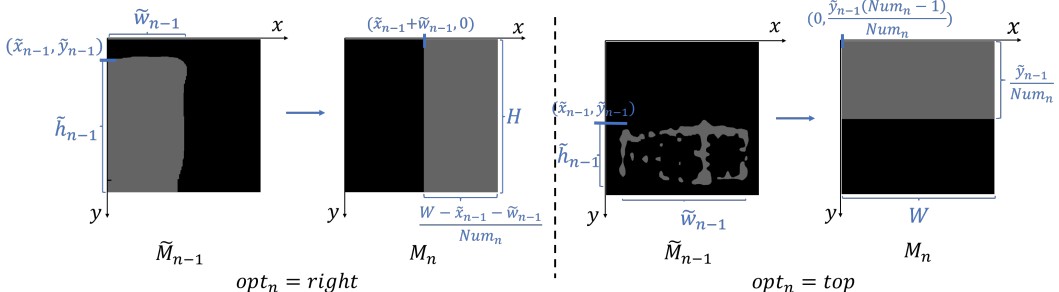

Figure 10: The rough mask $M_n$ of $\text{obj}_n(n > 1)$ is derived from the precise mask $\tilde{M}_{n-1}$ of the previously generated object $\text{obj}_{n-1}$.

## L   MORE DETAILS ON THE OVERLAPPING PROCESSING

Given $\text{opt}_n$ and $\tilde{M}_{n-1}$, the rough mask $M_{n,i}$ can be computed as

$$M_{n,i} = \begin{cases} \left(\tilde{x}_{n-1} \cdot r_i + (\tilde{x}_{n-1} + \tilde{w}_{n-1}) \cdot (1 - r_i), \tilde{y}_{n-1}, \tilde{w}_{n-1} \cdot r_i + \frac{W - \tilde{x}_{n-1} - \tilde{w}_{n-1}}{\text{Num}_n}, \tilde{h}_{n-1}\right), \\ \text{if } \text{opt}_n = \text{right}, \\ \\ \left(\tilde{x}_{n-1}, \frac{(\text{Num}_n - 1) \cdot \tilde{y}_{n-1}}{\text{Num}_n}, \tilde{w}_{n-1}, \tilde{h}_{n-1} \cdot r_i + \frac{\tilde{y}_{n-1}}{\text{Num}_n}\right), \\ \text{if } \text{opt}_n = \text{top}. \end{cases}$$

$$(7)$$

The illustration for different overlapping ratios is shown in Figure 11.

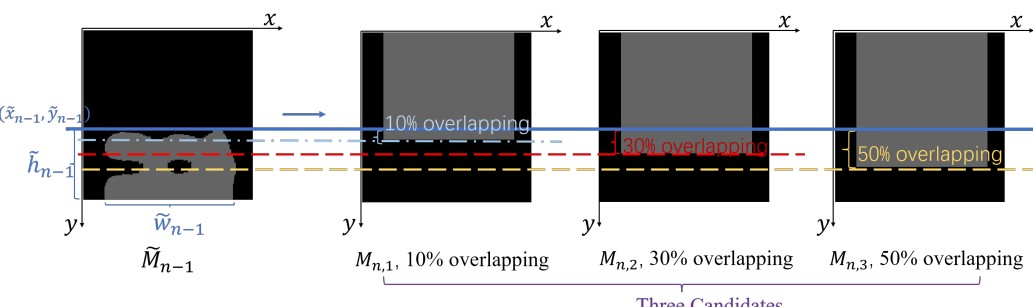

Figure 11: Three candidate masks $M_{n,i}$ of $\text{obj}_n$ at position $\text{opt}_n = \text{top}$. They correspond to $\text{obj}_n$ overlapping with 10%, 30%, and 50% of $\text{obj}_{n-1}$.

## M   MORE DETAILS ON THE EVALUATION QUESTIONNAIRE

As shown in Section 4, we design a questionnaire to comprehensively evaluate the alignment between the generated image and the text by GPT-4V (OpenAI, 2023) and human, from three aspects

- object completeness, correctness of attribute bindings, and correctness of spatial relationships. Specifically, given an image and a text prompt, for object completeness, we will evaluate if the image contains each single object in the prompt. If the object appears in the image, we will then judge if the attribute bindings of the object in the image align with the corresponding attribute bindings in the text prompt, to evaluate the correctness of attribute bindings. We will also ask GPT-4V or human to judge if the spatial relationships are correct and match the text, as the evaluation of the spatial relationships.

Examples of the questionnaire for different images and text prompts are shown in Figure 12.

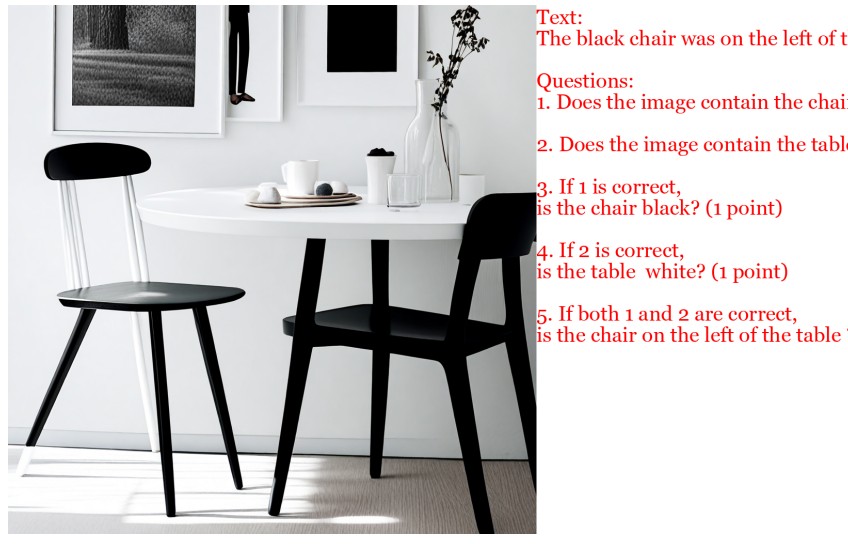

Text:
The black chair was on the left of the white table

Questions:
1. Does the image contain the chair? (1 point)

2. Does the image contain the table ? (1 point)

3. If 1 is correct,
is the chair black? (1 point)

4. If 2 is correct,
is the table white? (1 point)

5. If both 1 and 2 are correct,
is the chair on the left of the table ? (1 point.)

(a)

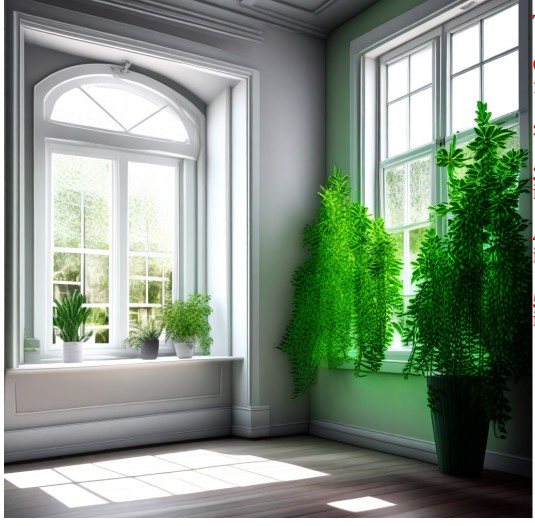

Text:
The green plant was on the right of the white window

Questions:
1. Does the image contain the plant? (1 point)

2. Does the image contain the window ? (1 point)

3. If 1 is correct,
is the plant green? (1 point)

4. If 2 is correct,
is the window white? (1 point)

5. If both 1 and 2 are correct,
is the plant on the right of the window ? (1 point.)

(b)

Figure 12: Illustration of the questionnaire for the evaluation of generated images

## N    LIMITATIONS

**Inference time of MuLan**    Since MuLan generates objects in a progressive manner, it will take longer time than one-stage methods. However, there is a tradeoff between accuracy and efficiency. Most existing one-stage methods generally fail on the complex prompts we focus on. We aim to

accurately and precisely control the generation process by the proposed progressive pipeline. To show the tradeoff more clearly, we conducted experimental comparisons on how the image-prompt alignment and inference time would vary with the increasing number of objects. As shown in the visualization results of Figure 13, although the inference time of MuLan increases with more objects, the image-prompt alignment can be maintained. In one stage methods (e.g., SDXL (Podell et al., 2023), PixArt-$\alpha$ (Chen et al., 2023)), however, the alignment with prompt becomes worse and worse with more objects.

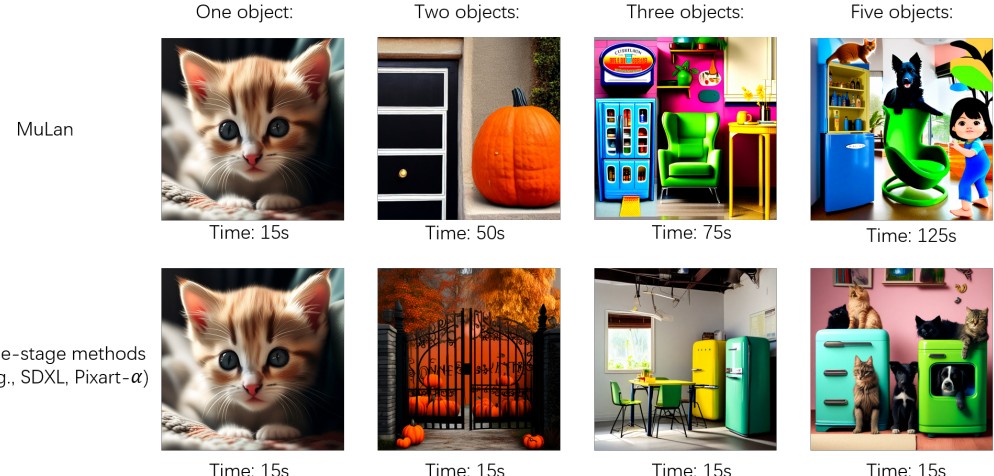

Figure 13: The inference time of MuLan and one-stage methods. The **prompts** are **'a cute kitten'**, **'the orange pumpkin is on the right of the black door'**, **'A blue refrigerator on the left, a green chair on the middle, and a yellow table on the right'**, and **'From left to right, an indoor room with a cute kitten sitting on top of a blue fridge, a black dog sitting on top of a green chair, and a cute kid'**, respectively. For one object, MuLan reduces to the utilized base diffusion model(e.g., SDXL (Podell et al., 2023)). For two or more objects, although MuLan requires more inference time, the image-prompt alignment can be maintained and controlled. This is a tradeoff between accuracy and efficiency. One-stage methods, however, generate worse and worse results with increasing objects.

Also, the inference time of MuLan is not linearly increasing with the number of objects. If the base model used in MuLan is powerful enough, several objects can be generated simultaneously in one stage, further reducing the inference time.

**Possible failure cases** Note that since MuLan is totally training-free, the generation capability highly depends on the off-the-shelf base model such as stable diffusion in MuLan. We discuss two more cases here. First, for those non-common single object that base model itself cannot generate, it is hard for base models to generate even a single object. In this case, MuLan also cannot generate correct objects. Secondly, for those non-common corner cases of image composition, such as the prompt 'in a bathroom, a huge dinosaur is sitting in a sink', MuLan may also fail to correctly generate them, as shown in Figure 14. The reason may be that for these cases, diffusion models cannot figure out reasonable relative size and practical scenes for them.

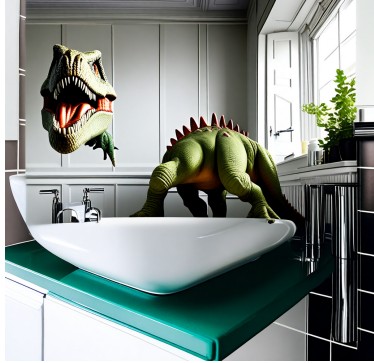

Figure 14: Possible failure case. In some non-common corner cases of image composition, like 'in a bathroom, a dinosaur is sitting in a sink', base diffusion models may fail to figure out relative size and practical scenes of objects, making generated images unnatural, as shown in the figure.

## O  MORE QUALITATIVE RESULTS

We show more examples of different methods in Figure 15.

Figure 15: More qualitative examples of images generated by different methods on intricate prompts.

