# OpenReview forum: "MuLan: Multimodal-LLM Agent for Progressive and Interactive Multi-Object Diffusion"
_ICLR.cc/2025/Conference — Submitted to ICLR 2025_

### Official Review · Reviewer_w3tY · 2024-11-03

**Soundness:** 2
**Presentation:** 3
**Contribution:** 2
**Rating:** 3
**Confidence:** 4

**Summary:**

This paper introduces MuLan, a comprehensive image generation method that utilizes a Large Language Model (LLM) agent for precise control of the generation process. The approach involves decomposing the prompt into a sequence of sub-tasks and generating each object sequentially through a diffusion model. Consequently, the method effectively generates multiple objects in accordance with the prompt.

**Strengths:**

- The idea of using Large Language Models (LLMs) for planning and Vision-Language Models (VLMs) to provide feedback is quite sensible.

- This approach allows for the generation of objects that closely adhere to given instructions.

**Weaknesses:**

1. Using LLMs as planners is not a novel concept. Several methods like RPG have explored this approach before.

2. In the experimental section, no compared methods leverage LLMs for image planning, although similar methods have been proposed. Only plain text-to-image methods are compared.

3. The entire generation process could be lengthy since each object in the image must be generated in order.

**Questions:**

1. The method is designed to generate objects progressively rather than all at once, but there is no ablation study demonstrating the benefits of this approach.

2. Additional baselines need to be compared, particularly those using large language models (LLMs) as planners.

3. How can we enhance the image quality of the generated outputs? While they successfully follow instructions, the resulting images don't appear to be as appealing as those produced by the base models.

---

> ### Author Response · Authors · 2024-11-26
> **Response to Reviewer w3tY**
>
> Thanks for your feedback. We have addressed your concerns in the following.
>
> **Q: Using LLMs as planners is not a novel concept. Several methods like RPG have explored this approach before.**
>
> **A:** We would like to clarify that MuLan is a concurrent work with RPG. MuLan was finished in January 2024 and first submitted to ICML2024, which is the same as RPG. We think we should not be penalized in terms of novelty by a concurrent work. Moreover, as we have elaborately discussed in the original manuscript, there are still substantial differences between MuLan and other methods using LLM as the planner (e.g., RPG).
>
> **Q: In the experimental section, no compared methods leverage LLMs for image planning, although similar methods have been proposed. Only plain text-to-image methods are compared.**
>
> **A:** As we have clarified, RPG is a concurrent work. To further show the effectiveness of MuLan, we have included more comparisons with powerful controllable/compositional image generation methods, like Ranni[1] and Composable Diffusion[2]. Please see **Table 6 in Appendix C** in the revised manuscript for details.
>
> **Q: The entire generation process could be lengthy since each object in the image must be generated in order.**
>
> **A:** Please note that there is a **tradeoff between accuracy and efficiency**. Most existing one-stage methods generally fail on the complex prompts we focus on. We aim to accurately and precisely control the generation process by the proposed progressive pipeline. To show the tradeoff more clearly, we have included the experimental comparisons on how the image-prompt alignment and inference time would vary with the increasing number of objects in Appendix N in the original manuscript. As shown in **Figure 13 in Appendix N**, although the inference time of MuLan increases with more objects, the image-prompt alignment can be maintained. In one stage methods (e.g., SDXL, PixArt-$\alpha$), however, the alignment with prompt becomes worse and worse with more objects.
> Also, the inference time of MuLan is not linearly increasing with the number of objects. If the base model used in MuLan is powerful enough, several objects can be generated simultaneously in one stage, further reducing the inference time.
>
> **Q: The method is designed to generate objects progressively rather than all at once, but there is no ablation study demonstrating the benefits of this approach.**
>
> **A:** Please note that the pipeline can only work in a progressive manner. Each object is generated conditioned on previously generated objects. It cannot generate all objects simultaneously. We design the progressive framework in this way to ensure that each object can be generated correctly.
>
> **Q: Additional baselines need to be compared, particularly those using large language models (LLMs) as planners.**
>
> **A:** We have included more comparisons with powerful compositional image generation methods, such as Ranni[1] and Composable Diffusion[2]. Please see Table 6 in Appendix C for details.
>
> **Q: How can we enhance the image quality of the generated outputs? While they successfully follow instructions, the resulting images don't appear to be as appealing as those produced by the base models.**
>
> **A:** Please note that since MuLan is a training-free method, it does not degrade the performance and image quality of utilized base models. To further show this, we have included more visualization results comparing MuLan with SDXL to the original SDXL in **Figure 7 in Appendix E** in the revised manuscript, demonstrating similar image quality of MuLan and SDXL. Hence, to enhance image quality, more powerful base models can be used during the generation.
>
> [1] Feng, Yutong, et al. "Ranni: Taming text-to-image diffusion for accurate instruction following." Proceedings of the IEEE/CVF Conference on Computer Vision and Pattern Recognition. 2024.
>
> [2] Liu, Nan, et al. "Compositional visual generation with composable diffusion models." European Conference on Computer Vision. Cham: Springer Nature Switzerland, 2022.

---

### Official Review · Reviewer_WGgb · 2024-11-03

**Soundness:** 3
**Presentation:** 1
**Contribution:** 2
**Rating:** 5
**Confidence:** 5

**Summary:**

This paper introduces MuLan (Multimodal-LLM Agent), which leverages the reasoning capabilities of Large Language Models (LLMs) to decompose complex prompts into multiple subtasks, progressively generating multi-object outputs with detailed planning and feedback control. Additionally, MuLan incorporates Vision Language Models (VLMs) to provide feedback, thereby enhancing the alignment between prompts and generated images. The authors conducted experiments with 200 prompts involving multi-object scenarios with complex relationships to evaluate MuLan, and the results demonstrate its superiority in generating multiple objects.

**Strengths:**

1. Utilize LLMs as planners and VLMs as inspectors to enhance generation in complex scenarios.

2. The approach is training-free and model-agnostic.

3. Qualitative results surpass those of SDXL and PixArt-α.

4. Supports human interaction throughout the generation process.

**Weaknesses:**

1. The results are not competitive enough compare to current open-source models like FLUX and SD3, the method are outdated and lack novelty .

2. As mentioned in L233-243, the rough mask is limited to just four relative positions, which restricts its ability to handle more complex scenarios and reduces its overall flexibility.

3. As mentioned in the limitations, Inference time of MuLan is much higher than base models, however, open-source models like sd3 could already achieve accurate generation in compositional scenarios. It is inefficient to use a mulit-step method which could not show superior advancement as presented in the paper.

**Questions:**

1. Since MuLan is a training-free framework, why don't you utilize SOTA models like FLUX or SD3, i would appreciate if you could provide more comparisons and results between SOTA models and your methods.

2. The authors deliberately stress the importance of dealing with overlapping problems, however, the paper do not present enough results about overlapping prompts, especially lacks the interactions between human and animals.  Can MuLan achieve accurate and harmony generation for more complex prompts with overlapping entities?

---

> ### Author Response · Authors · 2024-11-26
> **Response to Reviewer WGgb**
>
> Thanks for your feedback. We have addressed your concerns in the following.
>
> **Q: The results are not competitive enough compare to current open-source models like FLUX and SD3, the method are outdated and lack novelty.**
>
> **A:** We would like to clarify that this work was finished in January 2024 and firstly submitted to ICML2024. At that time, both SD3 and FLUX were not released. That is why we did not include the comparisons with them. Here, to show the effectiveness of MuLan, we have conducted experiments to qualitatively compare MuLan with SD3. Please see **Figure 6 in Appendix D** for details. The results show that SD3 still cannot deal with prompts with simple spatial relationships steadily, while MuLan can achieve better image-prompt alignment. Hence, the proposed framework is still valuable and meaningful.
>
> **Q: As mentioned in L233-243, the rough mask is limited to just four relative positions, which restricts its ability to handle more complex scenarios and reduces its overall flexibility.**
>
> **A:** Thanks for pointing out this. The reasons why only four positions are considered are as follows.
> 1. **We consider the similar setting about spatial relationships to T2ICompBench[1].**
>
>       In T2IBench, ‘left, right, top, bottom, near, next to, on the side of’ are considered. The relationships ‘near, next to, on the side of’ can be naturally contained in the previous four relationships and processed by MuLan.
>
> 2. **Most captions in coco can be described in these four relationships.**
>
>      For example, the captions contain ‘a large bus sitting next to a very tall building’, ‘Bunk bed with a narrow shelf sitting underneath it’, etc., which can be described by the four relationships.
>
> **Q: As mentioned in the limitations, Inference time of MuLan is much higher than base models, however, open-source models like sd3 could already achieve accurate generation in compositional scenarios. It is inefficient to use a mulit-step method which could not show superior advancement as presented in the paper.**
>
> **A:** As we have clarified, SD3 was released after we finished MuLan. On the other hand, as shown in Figure 6 in Appendix D, SD3 still cannot deal with prompts with spatial relationships well, while MuLan can achieve better image-prompt alignment and full control over the generation process. Hence, the proposed framework is still meaningful and may provide insights for future improvements.
>
> **Q: The authors deliberately stress the importance of dealing with overlapping problems, however, the paper do not present enough results about overlapping prompts, especially lacks the interactions between human and animals. Can MuLan achieve accurate and harmony generation for more complex prompts with overlapping entities?**
>
> **A:** Thanks for pointing out this. We have included more results on the performance of MuLan on complex overlapping prompts. Please see **Figure 8 in Appendix F** in the revised manuscript for details. The results show that MuLan can deal with different cases of complex overlapping prompts well.
>
> [1] Huang, Kaiyi, et al. "T2i-compbench: A comprehensive benchmark for open-world compositional text-to-image generation." Advances in Neural Information Processing Systems 36 (2023): 78723-78747.

---

### Official Review · Reviewer_T8AM · 2024-11-04

**Soundness:** 3
**Presentation:** 3
**Contribution:** 2
**Rating:** 3
**Confidence:** 3

**Summary:**

The paper introduces MuLan, a Multimodal-LLM agent to improve the performances of existing text-to-image generation models, especiallly with multiple objects, spatial relationships and attribute bindings.
The main contributions inlcude,
* A large language model (LLM) is adopted to decompose complex prompts into a sequence of simpler sub-tasks, each focusing on generating a single object.
* A vision-language model (VLM) provides feedback to ensure that each object is generated accurately and aligns with the original prompt.

**Strengths:**

(1) This article adopts LLM to divides text-to-image generation into several steps, it addresses the limitations of existing models in handling multiple objects effectively. (2) The use of an VLM to provide feedback ensures that the generated images maintain consistency to the input prompt.

**Weaknesses:**

(1) In Section 3.4, the paper mentioned 'MuLan will adjust the backward guidance of the current stage to re-generate the object', but detailed adjustment algorithm or operation is not clearly explained.

(2) The evaluation is not sufficient, more existing works e.g. Ranni[1], Composable[2] should be included.

(3) The baseline models (e.g. SD1.4, SDXL) used in this paper are relatively weak, I highly doubt that if MuLan still works when using more strong base models (e.g. SD3, FLUX)?

(4) The tradeoff between accuracy and efficiency should be evaluated quantitatively, so that we can assess the practical values of this work.

[1] Feng Y, Gong B, Chen D, et al. Ranni: Taming text-to-image diffusion for accurate instruction following

[2] Liu N, Li S, Du Y, et al. Compositional visual generation with composable diffusion models

**Questions:**

See Weaknesses

---

> ### Author Response · Authors · 2024-11-26
> **Response to Reviewer T8AM**
>
> Thanks for your feedback. We have addressed your concerns in the following.
>
> **Q: In Section 3.4, the paper mentioned 'MuLan will adjust the backward guidance of the current stage to re-generate the object', but detailed adjustment algorithm or operation is not clearly explained.**
>
> **A:** Thanks for pointing out this. We explain the procedure more here and **have included the explanation in the revised manuscript (Section 3.4, marked in blue)**. If the VLM detects any errors in the generated object, MuLan will modify the hyperparameters of backward guidance to control the strength of the guidance. We empirically found that the errors are typically the size or the position of the generated object. For example, the object may be too large and outside the bounding box (the rough mask). Hence the guidance strength needs to be larger to correctly generate the object. In the whole generation process, if MuLan needs to regenerate an object, it will try different guidance strength, i.e., the weight of the gradient of the energy function (Equation 1), and the loss threshold that is used for stopping criteria of guidance. In cases with incorrect positions, it will also re-plan the spatial location and regenerate the object.
>
> **Q: The evaluation is not sufficient, more existing works e.g. Ranni[1], Composable[2] should be included.**
>
> **A:** We have included the results of Ranni and Composable in the revised manuscript. Please see **Table 6 in Appendix C** for details. The results show the proposed MuLan can still outperform these methods by a large margin.
>
> **Q: The baseline models (e.g. SD1.4, SDXL) used in this paper are relatively weak, I highly doubt that if MuLan still works when using more strong base models (e.g. SD3, FLUX)?**
>
> **A:** We would like to clarify that this work was finished in January 2024 and firstly submitted to ICML2024. At that time, both SD3 and FLUX were not released. That is why we did not include the comparisons with them. Here, to show the effectiveness of MuLan, we have conducted experiments to qualitatively compare MuLan with SD3. Please see **Figure 6 in Appendix D** for details. The results show that SD3 still cannot deal with prompts with simple spatial relationships steadily, while MuLan can achieve better image-prompt alignment.
>
> **Q: The tradeoff between accuracy and efficiency should be evaluated quantitatively, so that we can assess the practical values of this work.**
>
> **A:** Please note that most existing one-stage methods generally fail on the complex prompts we focus on. We aim to accurately and precisely control the generation process by the proposed progressive pipeline. There is a **tradeoff between accuracy and efficiency**. To show the tradeoff more clearly, we have included the experimental comparisons on how the image-prompt alignment and inference time would vary with the increasing number of objects in Appendix N in the original manuscript. As shown in **Figure 13 in Appendix N**, although the inference time of MuLan increases with more objects, the image-prompt alignment can be maintained. In one stage methods (e.g., SDXL, PixArt-$\alpha$), however, the alignment with prompt becomes worse and worse with more objects.
>
> Also, the inference time of MuLan is not linearly increasing with the number of objects. If the base model used in MuLan is powerful enough, several objects can be generated simultaneously in one stage, further reducing the inference time.

---

### Official Review · Reviewer_uaQp · 2024-11-07

**Soundness:** 2
**Presentation:** 3
**Contribution:** 2
**Rating:** 5
**Confidence:** 4

**Summary:**

The paper presents MuLan, a training-free multimodal language model agent designed to enhance text-to-image (T2I) generation. MuLan addresses challenges in generating images with multiple objects, focusing specifically on controlling spatial relationships, relative sizes, and attribute bindings. By leveraging a large language model (LLM) for planning and a vision-language model (VLM) for feedback, MuLan decomposes complex prompts into sequential subtasks, each handling a single object generation with attention-guided positioning.

**Strengths:**

+ The paper is well-written and easy to follow.
+ MuLan demonstrates good control over the generation process and produces high-quality images that align with the prompts.
+ MuLan can be applied to human-agent interaction during the generation process.

**Weaknesses:**

- MuLan increases inference time, especially as the number of objects in a prompt grows, which could limit its scalability in real-time applications.
- As a training-free approach, MuLan is heavily reliant on the capabilities of underlying base models (such as Stable Diffusion).
- In some cases, as shown in Figure 2, the generated images exhibit unrealistic proportions. For example, in the first row, the refrigerator, chair, and table are the same size, and in the second row, the pumpkin and door are also similarly sized, which detracts from the realism of the generated scenes.
- Although qualitative results are emphasized, the absence of metrics such as generation speed or quantitative latency comparisons with baselines makes it difficult to assess MuLan’s practical efficiency.

**Questions:**

1. How efficient is MuLan, particularly as the number of objects increases in the prompt?
2. If the generated image deviates from the original prompt, how many iterations does MuLan typically require to produce an accurate result?
3. Could you provide more details on MuLan’s performance in handling edge cases, such as generating scenes with objects that have extreme relative sizes or complex occlusions?

---

> ### Author Response · Authors · 2024-11-26
> **Response to Reviewer uaQp (1/2)**
>
> Thanks for your feedback. We have addressed your concerns in the following.
>
> **Q: MuLan increases inference time, especially as the number of objects in a prompt grows, which could limit its scalability in real-time applications.**
>
> **A:** Please note that there is a **tradeoff between accuracy and efficiency**. Most existing one-stage methods generally fail on the complex prompts we focus on. We aim to accurately and precisely control the generation process by the proposed progressive pipeline. To show the tradeoff more clearly, we have included the experimental comparisons on how the image-prompt alignment and inference time would vary with the increasing number of objects in Appendix N in the original manuscript. As shown in **Figure 13 in Appendix N**, although the inference time of MuLan increases with more objects, the image-prompt alignment can be maintained. In one stage methods (e.g., SDXL, PixArt-$\alpha$), however, the alignment with prompt becomes worse and worse with more objects.
>
> Also, the inference time of MuLan is not linearly increasing with the number of objects. If the base model used in MuLan is powerful enough, several objects can be generated simultaneously in one stage, further reducing the inference time.
>
> **Q: As a training-free approach, MuLan is heavily reliant on the capabilities of underlying base models (such as Stable Diffusion).**
>
> **A:** Please note that the visual quality (e.g., the realism) of generated images indeed depends on the base models. Better base models can generate more realistic images. However, the main focus of MuLan is to improve the image-prompt alignment and achieve controllable generation by the proposed progressive framework instead of the quality itself. MuLan can be incorporated into any models and boost the performance of image-prompt alignment of base models. On the other hand, training-based methods for controllable generation require construction of large-scale dataset which may not be feasible, and are not as flexible as training-free methods are.
>
> **Q: In some cases, as shown in Figure 2, the generated images exhibit unrealistic proportions. For example, in the first row, the refrigerator, chair, and table are the same size, and in the second row, the pumpkin and door are also similarly sized, which detracts from the realism of the generated scenes.**
>
> **A:** We would like to clarify that MuLan **would not** degrade the realism of generated images compared to utilized base models. Since MuLan is training-free, the realism of generated images depends on the base models. To further show this, we include more visualization results comparing the realism between MuLan with SDXL and the original SDXL. Please see **Appendix E in the revised manuscript**. Moreover, as we illustrated in the human-agent interaction section (Fig. 4), users can easily adjust the generated images (e.g., size, shape, color, etc.) in practice, making the framework more flexible.
>
> **Q: Although qualitative results are emphasized, the absence of metrics such as generation speed or quantitative latency comparisons with baselines makes it difficult to assess MuLan’s practical efficiency.**
>
> **A:** We have included quantitative comparison of inference speed between MuLan and other one-stage methods in the original manuscript. Please see Figure 13 in Appendix N for details. Although the inference time of MuLan increases as the number of objects increases, it can achieve much better image-prompt alignment.
>
> **Q: How efficient is MuLan, particularly as the number of objects increases in the prompt?**
>
> **A:** We have included quantitative comparison of inference speed between MuLan and other one-stage methods in the original manuscript. Please see Figure 13 in Appendix N for details. Although the inference time of MuLan increases as the number of objects increases, it can achieve much better image-prompt alignment.

---

> ### Author Response · Authors · 2024-11-26
> **Response to Reviewer uaQp (2/2)**
>
> **Q: If the generated image deviates from the original prompt, how many iterations does MuLan typically require to produce an accurate result?**
>
> **A:** Generally it depends on the objects to be generated and the hyperparameters of backward guidance. If the generation is incorrect, MuLan would adjust the parameters of backward guidance to regenerate the objects. In general, this would take two or three iterations to generate correct objects. We also **explain more on how MuLan re-generates the objects in Section 3.4 (marked in blue) in the revised manuscript**. Please check it for details.
>
> **Q: Could you provide more details on MuLan’s performance in handling edge cases, such as generating scenes with objects that have extreme relative sizes or complex occlusions?**
>
> **A:** We have discussed some corner cases in Appendix N in the original manuscript, which includes cases with extreme relative size. Please check that part. Moreover, we have included more visualization results on cases with complex occlusion in **Appendix F in the revised manuscript**. The results show that MuLan can deal with different cases with complex overlapping. Please check the results.

---

### Author Response · Authors · 2024-11-26
**Response to all Reviewers**

Firstly, we would like to express our gratitude to all of our reviewers for their efforts and attention to details in reviewing our paper. We are particularly encouraged by the positive feedbacks we received, such as:
1. Effective framework to address the existing limitations in the field (R1, R2)
2. Good presentation and easy to follow (R1, R2, R4)
3. Adaptability to different applications (R1, R3)

Secondly, we are also blessed with many creative suggestions that our reviewers provided, and we have incorporated them into our revision.

Lastly, our reviewers expressed some concerns in the weakness section, and many of the concerns are, in fact, asking for clarification. We will address every point mentioned individually below, and we hope our response can help you in finalizing the scores of our paper. If you have any other questions, please feel free to reply back, and we will answer them asap!

**Highlighted changes in the revision (all revised parts are marked in blue):**
1. [TEXT] + More explanation on the adjustment process
2. [EXP] + More comparisons with latest controllable image generation methods
3. [EXP] + More comparisons with latest general image generation models
4. [EXP] + More results on complex prompts with overlapping

Sincerely,

MuLan authors

---

### Meta-Review · Area_Chair_7geY · 2024-12-16

**Metareview:**

The reviewers are concerned about the insufficient experimental comparison, unreaching sota performance, and limited technology. While the authors try to address these issues by providing specific explanations, the overall contribution remains limited. Overall, the ac has checked all the files and stands on the reviewers' side. The authors are suggested to further improve the current submission.

**Additional Comments On Reviewer Discussion:**

The reviewer [uaQp] raised weakness issues regarding the inference time increasing and SD model reliance, which are not well addressed by the authors. Also, the technical novel issue raised by [w3tY] is still not convincingly addressed by the authors.

---

### Decision · Program_Chairs · 2025-01-22

Reject